# A sensitive high repetition rate arrival time monitor for X-ray free electron lasers

Michael Diez [1,2] ✉, Henning Kirchberg[3,4], Andreas Galler[1] ✉,
Sebastian Schulz [5], Mykola Biednov[1], Christina Bömer[1,5], Tae-Kyu Choi [1,6],
Angel Rodriguez-Fernandez [1], Wojciech Gawelda [1,7,8,9], Dmitry Khakhulin[1],
Katharina Kubicek[1,2,10], Frederico Lima[1], Florian Otte [1], Peter Zalden[1],
Ryan Coffee [11,12], Michael Thorwart [2,3] & Christian Bressler [1,2,10] ✉

X-ray free-electron laser sources enable time-resolved X-ray studies with unmatched temporal resolution. To fully exploit ultrashort X-ray pulses, timing tools are essential. However, new high repetition rate X-ray facilities present challenges for currently used timing tool schemes. Here we address this issue by demonstrating a sensitive timing tool scheme to enhance experimental time resolution in pump-probe experiments at very high pulse repetition rates. Our method employs a self-referenced detection scheme using a time-sheared chirped optical pulse traversing an X-ray stimulated diamond plate. By formulating an effective medium theory, we confirm subtle refractive index changes, induced by sub-milli-Joule intense X-ray pulses, that are measured in our experiment. The system utilizes a Common-Path-Interferometer to detect X-ray-induced phase shifts of the optical probe pulse transmitted through the diamond sample. Owing to the thermal stability of diamond, our approach is well-suited for MHz pulse repetition rates in superconducting linear accelerator-based free-electron lasers.

In the last decade, the advent of intense femtosecond (fs) X-ray Free Electron Lasers (XFELs) launched the need for accurate control of the timing between two independent light sources: a free electron laser starts off with the photoinjector gun on the order of one kilometer away from the experiment, and a femtosecond optical laser source, typically located in the vicinity of a few tens of meters around the experiment. For femtosecond pump-probe X-ray experiments it is required to have accurate control of the timing. A precise synchronization scheme between both independent light sources is required for sub 100 fs timing jitter between both sources. To further increase the time resolution, the relative arrival time of the X-ray and optical laser pulses are measured close to the experiment with an appropriate diagnostic. Over the past decade, two main techniques have matured for reliable use with hard X-ray photon energies at different XFEL facilities, known by their brief names spectral[1–3] and spatial[4–6] encoding schemes. These schemes have been established at low repetition rate XFEL sources. At high repetition rate machines, such as European XFEL (EuXFEL)[7,8], the MHz pulse sequence requires sample relaxation time scales well below a few hundred nanoseconds, and only one study attempted to enter this range[9]. In addition, the deposited energy by a

[1]European XFEL GmbH, Holzkoppel 4, 22869 Schenefeld, Germany. [2]The Hamburg Centre for Ultrafast Imaging, Luruper Chaussee 149, 22761 Hamburg, Germany. [3]I. Institut für Theoretische Physik, Universität Hamburg, Notkestr. 9, 22607 Hamburg, Germany. [4]Department of Chemistry, University of Pennsylvania, Philadelphia, Pennsylvania 19104, USA. [5]Deutsches Elektronen-Synchrotron DESY, Notkestraße 85, 22607 Hamburg, Germany. [6]XFEL division, Pohang Accelerator Laboratory, Jigok-ro 127-80, 37673 Pohang, Republic of Korea. [7]Faculty of Physics, Adam Mickiewicz University, ul. Uniwersytetu Poznańskiego 2, 61-614 Poznań, Poland. [8]Department of Chemistry, Faculty of Sciences, Universidad Autónoma de Madrid, 28049 Madrid, Spain. [9]IMDEA-Nanociencia, Calle Faraday 9, 28049 Madrid, Spain. [10]Fachbereich Physik, Universität Hamburg, Notkestraße 9-11, 22607 Hamburg, Germany. [11]SLAC National Accelerator Laboratory, 2575 Sand Hill Rd., Menlo Park, CA 94025, USA. [12]The Pulse Institute, SLAC National Accelerator Laboratory, Menlo Park, CA 94028, USA. ✉e-mail: michael.diez@xfel.eu; andreas.galler@xfel.eu; christian.bressler@xfel.eu

fully filled X-ray pulse-train heats up the sample, potentially destroying conventional timing-tool samples such as silicon nitride ($Si_3N_4$) or yttrium aluminium garnet (YAG) within a single pulse-train (Supplementary Note 1). This fact requires new timingtool schemes with interaction samples able to withstand the sequence of intense X-ray pulses at MHz repetition rates. Diamond is a prime candidate as interaction sample, but can not be used in conventional spectral or spatial encoding schemes due to very weak X-ray-induced optical absorption or transmission changes[3].

In this work we describe a self-referenced timing tool scheme with an increased sensitivity. This scheme could be applied at all X-ray wavelengths from soft (<1 keV) to very hard (>25 keV) X-ray energies. The self-referenced scheme is not only sensitive to the change of amplitude of the transmitted optical probe pulse of the timing tool, but is also sensitive to the altered phase of the optical pulse. This increased sensitivity is achieved by time-shearing the optical pulse in a Common-Path-Interferometer (CPI). The optical pump-probe laser can be synchronised to the X-ray pulses by using two different synchronisation schemes: The precise and state-of-the-art optical synchronization (OS)[10, 11] and the more versatile but less precise radiofrequency based synchronisation (RFS)[12]. We measure the timing jitter at the Femtosecond X-ray Experiments Instrument (FXE)[13, 14] of EuXFEL, using both available synchronisation schemes. In addition, we can extract the X-ray-induced refractive index change with this timing tool method, to which the signal amplitude is directly correlated. While the Drude model is commonly used to describe the X-ray-induced optical changes in the sample[1, 6], we use the more advanced Maxwell-Garnett theory[15], an effective medium approximation for dielectric mixtures, to calculate the X-ray-induced refractive index change, and we show that this approach provides a more precise description than the commonly used Drude model.

## Results

### Experimental setup

The experiment was carried out at the FXE instrument at the end of the SASE1 photon beamline of EuXFEL at a fixed X-ray photon energy of 9.3 keV and a mean X-ray pulse energy of 300 $\mu J$. The pulse width is expected to be around 50 fs at full width at half maximum (FWHM)[8]. EuXFEL uses a superconducting linear accelerator, enabling the unique feature to produce trains of up to 2700 electron bunches within one 600 $\mu s$ long radiofrequency (RF) pulse[8]. With a 10 Hz RF pulse train repetition rate, EuXFEL is capable of generating X-ray pulses in a pulse-train pattern with an intra-train repetition rate of up to 4.5 MHz, which, together with the inter-train repetition rate of 10 Hz yields a total of up to 27.000 X-ray pulses per second. We used a train filling pattern at 1.128 MHz repetition rate and 100 X-ray pulses stored in each pulse train. The X-ray pulses were pre-collimated in the photon tunnel using

compound refractive lenses (CRLs) to match the aperture of downstream optics. Another stack of CRLs, located about 5 m upstream from the sample was used to achieve a nominal spot size of 150 ± 10 $\mu m$ at the sample position[13, 14], which matches the same expected size under conditions yielding a pre-focus inside the photon transport tunnel. The in-house developed EuXFEL optical pump-probe laser system was utilised, which is synchronised to the facility's main oscillator, and matches any chosen X-ray pulse pattern of the facility[16]. It delivers 15 fs (rms) ultrashort pulses at its 800 nm central wavelength. The 800 nm fundamental optical beam was frequency doubled with a $\beta$BBO (d = 0.5 mm $\theta$ = 29.2° and $\phi$ = 90°) crystal to generate optical pulses centered around 400 nm with a FWHM bandwidth of around 20 nm with a conversion efficiency of around 15%. The entire setup of the self-referenced timing tool was mounted on the multi-axis sample-stack, enabling the movement of the entire setup with all required degrees of freedom[13, 14].

The concept is based on an X-ray and optical cross-correlation scheme, using a frequency-to-time mapping to imprint the relative X-ray arrival time in a chirped optical pulse[1, 3]. The setup is based on a CPI, which creates a self-referenced background-free measurement (see Methods). An arrival time signal will only be measured in the presence of an X-ray pulse, otherwise, a zero-signal will be obtained. Our setup uses interferometric sensitivity to measure the phase- (and amplitude) change of a transmitted optical pulse after passing through a sample whose refractive index is changed by the partial absorption of a simultaneously transmitted X-ray pulse.

A schematic of the setup is shown in Fig. 1. The optical laser pulses are guided over a motorised optical delay stage $\Delta t_1$, which is used for temporal fine adjustment between the optical and X-ray pulses. The 800 nm pulses are then frequency doubled via second harmonic generation (SHG). For the relative arrival time measurement, a timing window of about 1 ps is needed to reliably cover the entire range of the temporal jitter between optical and X-ray pulses. We achieve this by chirping the optical pulses to the length of the desired timing window by transmitting the 400 nm pulse through a dispersive optical element (flint glass SF11) Disp., whose thickness is chosen to match the desired amount of chirped pulse length. The dispersive element orders the optical frequencies in time yielding smaller frequencies (or wavelengths on the red side) in front and higher frequencies (or wavelengths on the blue side) on the trailing side of the pulse.

The chirped pulses enter the main part of the timing tool, the CPI, consisting of two polarisers $P_1$ and $P_2$ before and after the sample and two 5mm thick a-cut $\alpha BBO_{1,2}$ crystals. The interaction sample S, where the X-ray and optical pulse pairs are temporally and spatially overlapped, is placed in the center of the CPI. The sample is used to imprint the X-ray arrival time by an X-ray-induced refractive index change into the simultaneously transmitted optical pulse. Two lenses are used to

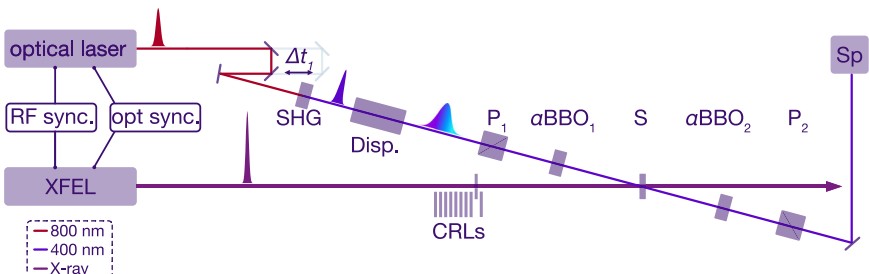

**Fig. 1 | Self-referenced timing tool setup.** An ultrashort 800 nm optical pulse is guided through a motorised delay stage $\Delta t_1$ and frequency doubled to 400 nm in a second harmonic generation (SHG) crystal. A dispersive flint glass block (Disp.) is used to further chirp the optical pulse duration. The chirped pulse is guided through the Common-Path-Interferometer (CPI) consisting of two polarisers $P_{1,2}$ and two a-cut $\alpha BBO_{1,2}$ crystals. The optical pulse and the X-ray pulse (XFEL) are temporally and spatially overlapped in the diamond sample S in the center of the CPI, where the X-ray arrival time is imprinted into the optical pulse. The spectrally encoded arrival time is analyzed with a spectrograph (Sp). The optical laser and XFEL sources are either synchronised with a radiofrequency based sychronisation (RF sync.) or an optical synchronsiation scheme (opt. sync.). The X-ray beam is focused by using a compound refractive lense stack (CRLs).

focus the optical pulses onto the sample (40 μm FWHM) and to collimate the optical beam exiting the sample. The optical pulses were then guided to a spectrograph SP (Andor Shamrock 193i with a 600 l/mm grating) to evaluate the self-referenced X-ray arrival time signal. Various neutral density filters (Thorlabs ND Filter with AR coating) were placed in front of the entrance slit to protect the detector from optical saturation. When not stated otherwise, the standard configuration during this experiment used a 1.5 OD filter (Thorlabs NE15A-A) in front of the spectrometer. For the detection of the individual spectra, we used a Gotthard detector[17] in the focal plane of the spectrograph, synchronised to the optical source and operated at 564 kHz, thus, recording every second optical pulse within the pulse trains. In the near future, new strip detectors such as the Gotthard-II[18] and the KALYPSO[19] detectors can capture every single pulse of a full pulse train at 4.5 MHz repetition rate. Another possibility is to use a second Gotthard detector attached to a second output of the Andor spectrograph, such that these two Gotthard detectors can be operated to record the arrival time spectra for every pulse at 1.1 MHz in an alternating fashion. Fused-silica glass blocks with a thickness of 0.4 cm or 2.5 cm were added to the 400 nm optical beam path to chirp the optical pulses to further increase the temporal measurement window.

## X-ray-induced transient refractive index

The fundamental effect, which alters the refractive index upon impinging X-ray pulses, is the absorption of X-ray photons and subsequent generation of charge carriers in the sample material, which effectively increase the charge carrier density. The created charges change the band structure of the material, and, thus, affect the refractive index. To get an idea of the magnitude of the transient refractive index change, the X-ray-induced electron density after an X-ray pulse needs to be estimated.

An absorbed X-ray photon excites a core shell electron and ejects it from the atom, leaving a hole behind. On the fast time scales, the hole decays within femtoseconds via an Auger decay, while fluorescent decay processes happen on much longer time scales[20]. The fast Auger decay initiates an electron cascade through inelastic electron-electron collisions. This cascade will last until the cascading electrons reach an energy below their pair creation energy[21,22]. The time scale of the full electron cascade after an initial X-ray photon absorption depends on the X-ray photon energy. In general the electron cascade for low energetic X-ray photons is shorter (< 20 fs) due to lesser cascading steps needed, until the electron energy reaches the pair creation energy of 12.2 eV in diamond. For high-energy photons (> 10 keV), the cascading can reach time scales of 100 femtoseconds[23]. The so generated electrons in the conduction band have a long lifetime in the 1–3 ns range[24].

Simulated electron densities $N_e$ for three different X-ray beam diameters in a 50 μm diamond sample are shown in Fig. 2a–c. Further information of the calculation can be found in the Methods section. The simulation illustrates the expected charge carrier densities (cm⁻³) for a broad range of X-ray pulse energies (y-axis) over an X-ray photon energy range from 5 keV to 25 kev and three different X-ray FWHM beam diameters of 20 μm, 100 μm and 200 μm. From the previous section, it can be deduced that X-ray pulses generating an electron density above $1 \times 10^{21}$ electrons cm⁻³ are increasing the diamond temperature above its melting point. Therefore, these electron densities are ruled out in Fig. 2a.

## Drude model.

The optical properties of solids are closely related to the electronic properties of a material. The dielectric function $\epsilon(\omega)$ is influenced by free carriers. These free carriers are often responsible for ultrafast electrical changes in materials after irradiation with ultrashort optical or X-ray pulses[25]. A simple model to estimate the contribution of free electrons is a model based on classical conductivity, called the Drude model. This plasma model is based on a gas of free electrons. In

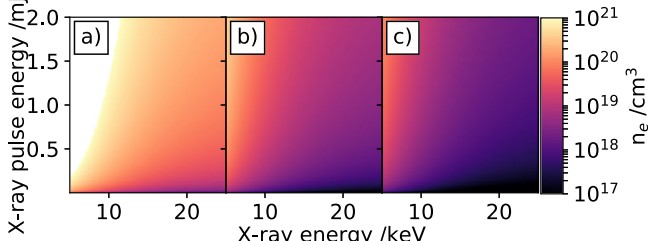

**Fig. 2 | Calculated X-Ray induced electron density.** Calculation of electron densities in a 50 μm thick diamond sample for different X-ray pulse energies over a wide range of X-ray photon energies. The carrier densities are calculated for three different full-width at half maximum X-ray beam diameters: (**a**) 20 μm, (**b**) 100 μm, and (**c**) 200 μm. For details see text.

this approach, details of the lattice potential and electron-electron interactions are not included. This simple model is frequently used to estimate the expected X-ray-induced refractive index changes in various materials[1,6]. The electric properties of a solid material are closely related to the real and imaginary parts of the dielectric function $\epsilon(\omega) = \epsilon_1(\omega) + i\epsilon_2(\omega)$. The dielectric function is dominated by the free electron density $N_e$. In the Drude model, the dielectric function is described in ref. [25], [26]

$$\epsilon_D(\omega) = \epsilon_r - \left(\frac{\omega_p}{\omega_o}\right)^2 \frac{1}{1 + i/\omega\tau} \tag{1}$$

where $\epsilon_r$ is the intrinsic dielectric constant, $\omega_p = \sqrt{N_e e^2/\epsilon_0 m_e^*}$ the plasma frequency, $\omega_o$ the central angular frequency of the optical laser and $\tau$ the mean free electron time between collisions. Here, $\epsilon_0$ is the vacuum permittivity and $m_e^*$ is the effective electron mass. The complex refractive index is defined as $n(\omega)^2 = \epsilon(\omega)$, thus the real and imaginary part of the refractive index can be calculated from the Drude model (see Methods).

**Maxwell-Garnett theory.** The change of the number of electrons in the conduction band alters the dielectric properties of diamond. The Drude model in Eq. (1) treats the free charge carriers as exponentially damped harmonic oscillators with a characteristic plasma frequency $\omega_p$ and damping being related to the mean free time between electron collisions[27]. A possible extension to this simple independent-particle model is to include the coupling of the induced free charge carriers with their dielectric background. The free electrons created by the incoming X-ray pulses interact instantaneously (i.e. within 0.3 fs)[28] with the bound electrons in the diamond crystal, distort their equilibrium distribution and thus polarise the diamond lattice. This self-induced electronic polarisation cloud screens the free charges, moves with them along the crystal and the electrons plus the distortion of the bound charges in the lattice form together an electronic polaron[29–31]. The dielectric properties of this quasiparticle differ slightly from those of the bare diamond lattice. We may then describe the diamond crystal in the presence of the polarons as a mixture of two different but isotropic dielectrics. The Maxwell-Garnett theory describes such dielectric mixtures with an effective combined dielectric function[32]

$$\epsilon_{MG}(\omega) = \epsilon_d(\omega) \frac{[2\epsilon_d(\omega) + \epsilon_p(\omega)] - 2l[\epsilon_d(\omega) - \epsilon_p(\omega)]}{[2\epsilon_d(\omega) + \epsilon_p(\omega)] + l[\epsilon_d(\omega) - \epsilon_p(\omega)]}, \tag{2}$$

where $\epsilon_d(\omega)$ and $\epsilon_p(\omega)$ are the dielectric functions of the bare diamond crystal and that created by the polaron, respectively. The volume fraction of the polarons $l = 4\pi N_e r_p^3/3$ depends on their radius $r_p$ and the free charge carrier density $N_e$. We use a Debye-type dielectric function $\epsilon_x(\omega) = \epsilon_{\infty,x} + (\epsilon_{S,x} - \epsilon_{\infty,x})/(1 - i\omega\tau_e D, x)$ with the respective high-($\epsilon_{\infty,x}$) and low-($\epsilon_{S,x}$) frequency dielectric constant and a specific Debye

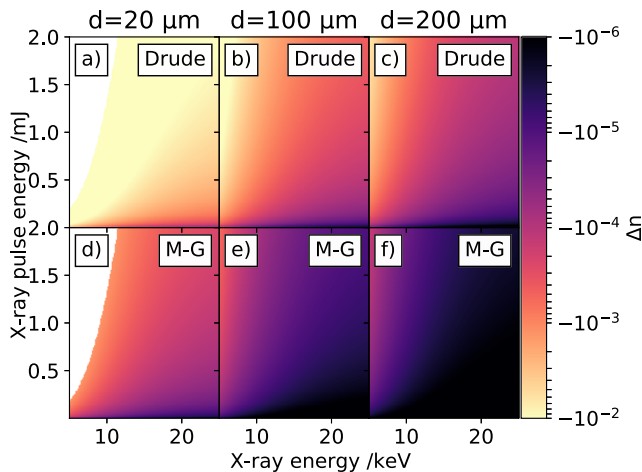

Fig. 3 | **Calculated X-ray induced refractive index change.** Change of the refractive index in a 50 $\mu m$ thick diamond sample for different X-ray pulse energies of a broad range of X-ray photon energies. Subfigures (**a**–**c**) show the expected refractive index change for X-ray pulses with a full-width at half maximum beam diameters of $d = 20\,\mu m$, $d = 100\,\mu m$ and $d = 200\,\mu m$, calculated with the Drude model. Correspondingly (**d**–**f**) illustrate the expected refractive index changes calculated with the Maxwell-Garnett (M-G) theory. For details see text.

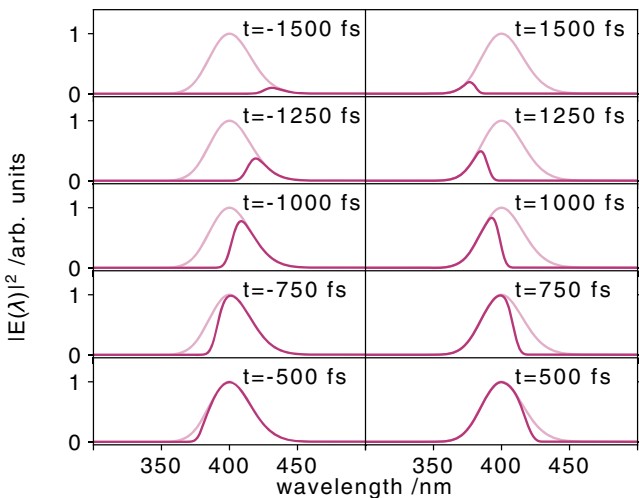

Fig. 4 | **Calculated self-referenced arrival time spectra.** Simulated self-referenced arrival time spectra for different X-ray arrival times from −1500 fs to 1500 fs (as indicated). The arrival time spectra are shown as solid purple lines, while the full original spectra of the time-sheared pulse are indicated by the shaded purple lines. Due to the chirp of the optical pulse, the arrival time signals are not exactly symmetric for corresponding positive and negative arrival time.

relaxation time $\tau_{\mathrm{D,x}}$. The involved parameters are displayed in Tab. S2 in the Supplementary and characterise both dielectric contributions. We assume a slight reduction of the high-frequency and static dielectric constant for the polarons in comparison with the bulk dielectrics of the diamond. The change of the dielectric properties of diamond due to polaron formation is in line with the results from time-dependent density functional theory calculations of diamond at an energetically excited state[33]. The decrease of the dielectric parameters can be understood in terms of the reduced overall electronic polarisability of the electrons in the polarons because the free electrons interact with the increased localised charge distribution in the form of the polarization cloud in the lattice. This causes more rigid charges and thus a smaller polarisability. Also, the reduced polarisability causes an increased Debye relaxation time $\tau_{\mathrm{D,p}}$ of the polaron. Eq. (2) is the result of the Maxwell-Garnett model which considers the electronic polarons as an ensemble of homogeneous dielectric spherical particles of radius $r_{\mathrm{p}} \leq a$ comparable or less than the interatomic spacing $a$ of the lattice (see Tab. S2 in the Supplementary Note 5 for the described model)[31].

**Self-referenced signal simulation.** Using the dielectric properties for diamond (Tab. S1, S2, and S3 in the Supplementary Note 5) and the two models described above, we have simulated the expected X-ray-induced transient refractive index change in diamond. The estimated refractive index change for X-ray photon energies ranging from 5 keV to 25 keV and X-ray pulse energies ranging from 1 $\mu J$ to 2 mJ calculated with the Drude- and Maxwell-Garnett model are shown in Fig. 3a–c and d–f, respectively. The X-ray beam profile is assumed to be a flat-top beam profile with beam diameters of $d = 20\,\mu m$ (a) and d)), $d = 100\,\mu m$ (b) and e)) and $d = 200\,\mu m$ (c) and f)). A beam size of $d = 20\,\mu m$ would represent a timingtool operated very close to the X-ray focus, while the other two beam diameters of $d = 100\,\mu m$ and $d = 200\,\mu m$ are more relaxed conditions far away from the focal plane. The white parts in Fig. 3a and d are due to X-ray conditions, where the absorbed X-ray photons would heat the diamond above its melting point and are therefore omitted.

The expected timing signal caused by the transient refractive index change described by the Drude- and Maxwell-Garnett theory can be estimated by linear propagation of an arbitrary shaped laser pulse through the involved optical elements (Supplementary Note 2). The signal strength of the self-referenced timing tool in diamond is directly

correlated to the X-ray-induced refractive index change in the diamond sample. Since the optical light of the timing tool probe pulse does not exhibit any significant X-ray-induced absorption or reflection, the timing tool signal amplitude is solely caused by the X-ray-induced optical phase shift after the X-ray pulse striked the sample. The phase shift is defined by

$$\Delta\phi = \frac{2\pi\,\Delta n}{\lambda_0}\,d, \qquad (3)$$

where $\Delta n$ is the transient refractive index change and $\lambda_0$ the central wavelength of the optical laser pulse. Depending on the relative arrival time of the ultrashort X-ray pulse and the longer time-sheared optical pulse, the X-ray-induced phase-shift is imprinted at different wavelengths in the optical pulse. A simulation of the self-referenced timing tool signal for various different relative arrival times and using actual experimental conditions is shown in Fig. 4.

**Arrival time measurements at the FXE instrument**
A pixel (thus wavelength) to time calibration is required to map the recorded self-referenced arrival time spectra onto a time axis, enabling the evaluation of the relative X-ray arrival time. For this purpose the optical delay stage was scanned, while using the more precise optical synchronisation scheme (OS) of the facility. For each time delay step thousands of arrival time spectra were recorded. With the known step size of the motorised delay stage, a linear calibration factor of $28 \pm 5$ fs/pixel was extracted (Supplementary Note 3).

To analyze individual arrival times, the edge positions of the individual arrival time spectra are determined by fitting the recorded spectra with an error-function-like fitting function. This procedure is exemplarily shown in the Supplementary Information (Fig. S5 and S6). The fit results with the time calibration yield an overall timing accuracy of 19 fs in this experiment. This precision is governed by the utilised spectrometer resolution and can be increased by using a more dispersive, e.g., by about a factor of two when using a 1200 l/mm.

During the experiment, the X-ray beam pointing was not stable due to subtle mechanical vibrations on the mirror system (comprising three mirrors) in the photon delivery tunnel, located several hundreds of meters upstream from the sample position[34]. This leads to a fluctuating spatial overlap between the X-ray and optical pulses in the

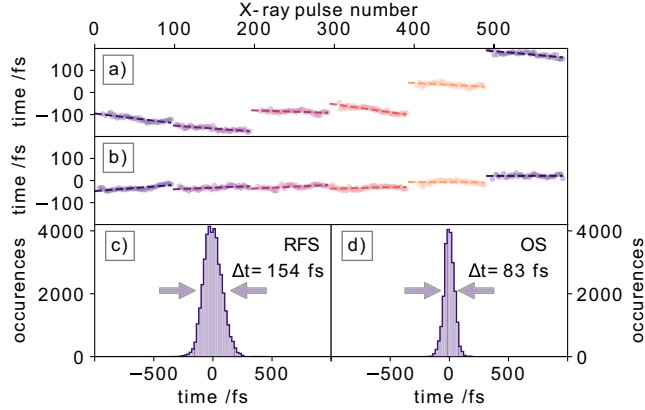

**Fig. 5 | Analysis of measured X-ray arrival times.** Relative arrival times for 6 consecutive pulse trains, with each 100 X-ray pulses, are shown, (**a**) via radio-frequency synchronisation (RFS) of the electron bunches, (**b**) via the optical synchronisation (OS) scheme. For each pulse train, a linear function is fitted to indicate the common drift patterns of the relative arrival times within a pulse train. The timing jitter distribution over a time span of 100 s is shown in (**c**) for the RFS ($\Delta t = 154 \pm 19$ fs) and for the OS ($\Delta t = 83 \pm 19$ fs) schemes.

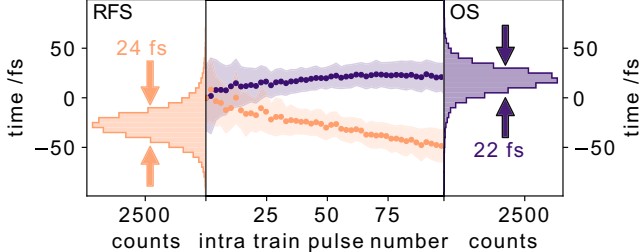

**Fig. 6 | Average intra-train arrival time drifts.** The average arrival times of the X-ray pulses within a pulse train are analysed in the center panel for both synchronisation schemes. X-ray pulses within a pulse train are drifting to earlier arrival times if the radiofrequency synchronisation scheme (RFS, orange) is used, while they are drifting to later arrival times within a pulse train for the optical synchronisation scheme (OS, blue). The intra-train timing distributions are shown on the left (RFS) and on the right (OS), yielding comparable full-width half maximum values when only using 100 pulses within a pulse train.

diamond sample affecting the signal quality. To reject spectra with bad spatial overlap between the two pulses, all self-referenced arrival time spectra with a signal-to-noise ratio below 2 were removed from the analysis. Analysis of the X-ray pointing jitter and the rejection of weak arrival time spectral intensities are given in reference[35].

The time arrival measurements were performed with both available facility synchronisation schemes, for which we determined the relative X-ray arrival time jitter (Fig. 5). Panels (a) (with the RFS scheme) and (b) (with the OS scheme) show the relative arrival times of every (second) individual X-ray pulse for six consecutive pulse trains. In this sequence we had x-ray pulse energies ranging from low to the highest value, which delivered an arrival time spectrum with S/N > 2. Figure 5c, d display the arrival time distribution for all available X-ray pulse arrival times within a 100 s time window. As expected, the RF-synchronisation shows a wider spread of the X-ray arrival=times, with $154 \pm 19$ fs FWHM (c) compared to the optical synchronisation with an arrival time distribution of $83 \pm 19$ fs FWHM (d). One should note that over a longer time periods, the RF-synchronisation timing jitter is expected to become worse, while the optical synchronisation timing jitter is expected to remain within the 83 fs FWHM window[9].

Interestingly, the pulse pattern in Fig. 5 between RF-synchronisation (a) and optical synchronisation (b) shows a different behavior for the relative arrival times within a single pulse train. As a guide to the eye, a linear function is fitted into each pulse train, indicating the trend of the drifting arrival times. Using the RF-synchronisation, the relative arrival times within a train generally drift to earlier arrival times, while for the OS scheme there is no clear trend observed within the pulse train.

Looking more closely into the intra-train drift pattern (Fig. 6) the linear decrease for the RFS synchronisation is displayed together with the nearly flat results using the OS scheme (central panel) for several thousand pulses (the first pulse of the pulse trains is set to 0 fs). The shaded area in Fig. 6 indicates the extracted standard deviation of the arrival times for each pulse number within a pulse train. Notably, the relative arrival times of the first 10 pulses within a pulse train are more uncertain (larger standard deviation) than the following X-ray pulses for either synchronisation scheme. The relative arrival time distribution for all measured pulse trains are shown on the left and right panels of Fig. 6 for RFS and OS schemes, respectively.

The FWHM of both distributions are very similar with 22 fs (OS) and 24 fs (RFS). Since both intra-train arrival time distributions are

nearly identical, it suggests that the optical synchronisation mainly stabilises the 10 Hz pulse-train to pulse-train timing jitter.

## X-ray-induced refractive index change

With the self-referenced timing tool we can directly probe the X-ray-induced transient refractive index change. To get an estimate of the X-ray-induced transient refractive index change, we simulate the self-referenced signal as described before with the actual experimental conditions and compare the simulation to the actual experiment. The X-ray-induced phase change, related to the transient refractive index via Eq. (3), is a free fitting parameter, which we use to scale the simulated signal amplitude to match the experimentally measured arrival time signal.

In order to perform the simulation, the full spectrum of the original laser pulse needs to be known. The total laser spectrum is measured by rotating the second polariser $P_2$, to fully transmit the original 45° polarised optical pulse. To protect the detector from over-saturation, a neutral density filter (Thorlabs NE20A-A) was placed in front of the spectrometer, transmitting 0.05% of the pulse at a central wavelength of 410 nm. In this configuration we recorded 1000 individual laser pulses and averaged all of them to obtain the spectrum of the laser pulse. This averaged spectrum will be used to reconstruct the electric field of the laser pulse for the simulation.

As described above and in more detail in the Supplementary Information (Supplementary Note 2), the obtained laser pulse from the actual experiment is linearly propagated through the setup. The goal of the simulation is to recreate the experimentally measured self-referenced arrival time spectra, using the known experimental conditions, where the only free parameter is the transient refractive index change. By using the transient refractive index as a fitting variable in the simulation, the amplitude of the simulated signal can be scaled to the measured arrival time signal. With this method we can deduce the magnitude of the X-ray-induced transient refractive index change in the measurement.

Due to the inevitable X-ray pointing instability, the arrival time spectra are subject to huge amplitude variations. This happens when the X-ray and optical pulses are not perfectly overlapped. Since the optical pulse focus size was set to be smaller than the X-ray focus size, we are only interested in the measurements with the highest self-referenced signal amplitudes. This ensures a perfect overlap between both pulses, where the optical pulse probed the maximal refractive index change in the center of the X-ray pulse footprint on the diamond sample. We use the highest 0.5% (1600 measurements) of the self-referenced spectra amplitudes to ensure that i) the X-ray pump and laser probe pulses fully overlap, and ii) a well-known (highest) X-ray pulse energy (here: 0.68 mJ) strike the sample. As described in the

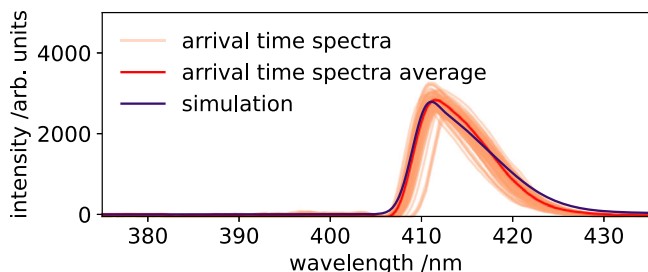

**Fig. 7 | Experimentally determined and simulated arrival time spectra.** Self-referenced arrival time spectra with the 0.5% highest amplitudes (orange). The average (red) of theses arrival time spectra is used as reference to fit a simulated (blue) arrival time spectrum to the data. The required X-ray-induced refractive index change to fit the simulation is $\Delta n_{\mathrm{exp}} = -5.7 \times 10^{-5}$. The horizontal shifts of the individual arrival time spectra are caused by the arrival time jitter, while the fluctuating amplitudes are caused by the X-ray pointing instabilities.

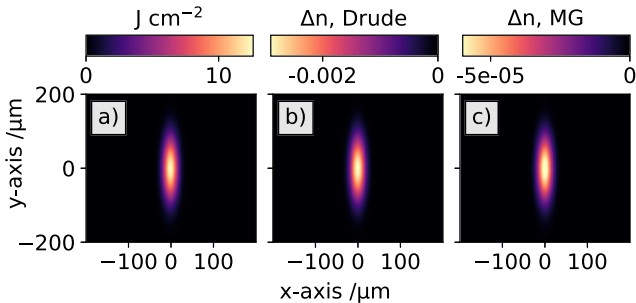

**Fig. 8 | Fluence-dependent refractive index change.** Analysis of the expected transient refractive index change with the experimentally used X-ray beam profile. The calculated fluence of the used X-ray beam is shown in (**a**). The expected transient refractive index changes in diamond calculated from the fluence are shown in (**b**) for the Drude model and (**c**) for the Maxwell-Garnett (MG) model. Please note the very different scales for each color bar.

experimental details section, the X-ray pulse energy at 9.3 keV was measured with an average pulse energy of $300\,\mu\mathrm{J}$. With the pulse resolved Intensity Position Monitor (IPM)[13], the average X-ray energy of the top 0.5% analysed data is determined with $640\,\mu\mathrm{J}$. Using the selected spectra, we analyze the X-ray-induced refractive index change, see Fig. 7. The selected self-referenced arrival time spectra are averaged and a simulated arrival time spectrum is fitted to match the experimental data. The X-ray-induced refractive index change to fit the experimental data is $\Delta n_{\mathrm{exp}} = -5.7 \times 10^{-5}$.

To calculate the expected X-ray-induced refractive index change with the Drude and Maxwell-Garnett models, the exact electron density generated by the X-ray-induced electron cascade needs to be known. Using the actual X-ray beam size at the sample position of $150 \pm 10 \times 30 \pm 2\,\mu\mathrm{m}$ and an X-ray pulse energy of $640\,\mu\mathrm{J}$, the transient refractive index in the diamond sample can be calculated as a function of the X-ray beam profile. The calculated results are shown in Fig. 8. The fluence dependent X-ray beam profile is depicted in subfigure a) with a maximum fluence in the center of the beam of $12\,\mathrm{J\,cm}^{-2}$. The calculated transient refractive index change in the $50\,\mu\mathrm{m}$ thick diamond sample is shown in subfigure b) for the Drude model and c) for the Maxwell-Garnett theory.

With the known conditions of the focal X-ray beamsize and X-ray pulse energy, the transient refractive index change in the center of the beam profile is calculated to be $\Delta n_{\mathrm{D}} = -2.9^{+0.1}_{-0.08} \times 10^{-3}$ and $\Delta n_{\mathrm{MG}} = -6.3^{+0.29}_{-0.26} \times 10^{-5}$ with the Drude and Maxwell-Garnett theory, respectively.

These calculated X-ray-induced refractive index changes are used to simulate the self-referenced arrival time signal and are compared to

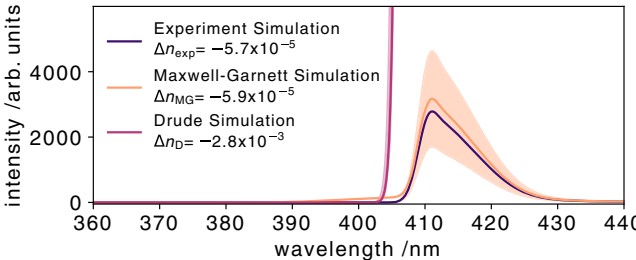

**Fig. 9 | Comparison of calculated arrival time spectra.** Estimation of the transient refractive index change in diamond in the actual experiment. The solid blue curve is a simulation which recreates the original experimental data with an X-ray-induced refractive index change of $\Delta n_{\mathrm{exp}} = -5.7 \times 10^{-5}$. The solid orange curve a numerical simulation. The orange curve is the simulated arrival time signal, using the Maxwell-Garnett ($\Delta n_{\mathrm{MG}}$) theory. The orange band gives the upper and lower expected transient refractive index change according to the Maxwell-Garnett model when implementing the uncertainty of the X-ray beam focal size of $150 \pm 20\,\mu\mathrm{m}$. The same applies to the purple curve which indicates the calculated values for the same conditions but with the Drude model ($\Delta n_{\mathrm{D}}$).

the simulation which reproduces the experimental data in Fig. 7. The widely used Drude model (purple) heavily overestimates the X-ray-induced refractive index change when compared with the experimental (blue) data, see Fig. 9. The simulated arrival time spectrum with the Maxwell-Garnett model (orange) and the experimentally measured arrival time signal are matching within the error bars (orange shaded area) of the simulation. The error bars of the simulation are due to the uncertainty of the X-ray focus size and the corresponding X-ray-induced electron density in the diamond sample.

With an experimentally determined X-ray-induced refractive index change of $-5.7 \times 10^{-5}$, the phase shift corresponds to 2.57°. The theoretically predicted phase shifts are 2.66° (Maxwell-Garnett) and 126° (Drude). The large predicted X-ray-induced phase shift by the Drude model would be clearly visible in the arrival time spectra as self-phase modulation like oscillations. Phase shifts larger than one cycle of the optical wavelength cause severe oscillations in the optical arrival time spectrum (Fig. S4 in the Supplementary Information). The non-existing observation of these oscillations in the measured optical arrival time spectra support our finding of a very small X-ray-induced refractive index change on only a few degrees and contradicts the larger phase shift predicted by the Drude model.

## Discussion
We have introduced a self-referenced detection scheme to measure the relative X-ray arrival time of hard X-ray pulses at XFEL facilities at MHz repetition rates. A major step towards future timing-tools is accomplished by developing a timing tool allowing the usage of diamond as interaction sample. With this we pave the way for timing tools which are nearly non-invasive compared to the other commonly used spectral and spatial encoding timing tool, due to the very low X-ray absorption cross-section of diamond in the hard X-ray regime. For future MHz repetition rate XFELs diamond might be the only material which is able to withstand the operation with intense X-ray pulses. In addition, the presented method possesses some advantages over other commonly used timing tool schemes. It is easier to set up and operate than a THz-Streaking experiment[36] and it is more sensitive and less invasive than the established spatial- and spectral encoding schemes.

The X-ray arrival time jitter at the FXE instrument of EuXFEL was measured using a $50\,\mu\mathrm{m}$ thick diamond sample as interaction sample with the self-referenced detection scheme. Both facility synchronisation schemes were used and relative arrival time jitter of 154 fs FWHM and 83 fs FWHM were measured using the RFS and OS, respectively. The intra-train timing jitter is almost identical for both synchronisation

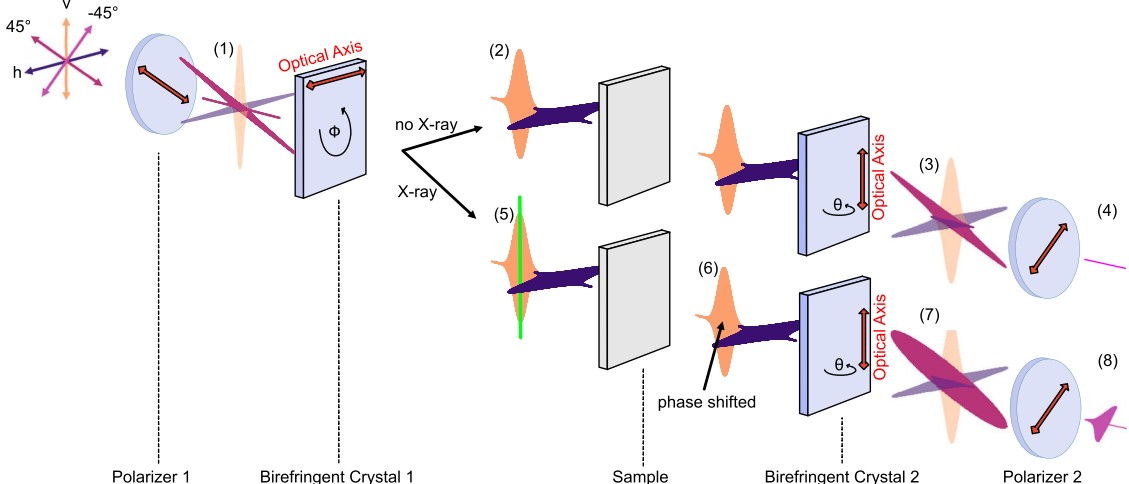

**Fig. 10 | Common-path-interferometer principle.** The first polariser defines the polarisation of the optical pulse to 45°, which can equally be described by two perpendicular polarisation components (PCs, 1). No phase-shift is introduced in the absence of any X-ray pulse and the two time-delayed PCs (2), generated by the first birefringent crystal (BC), can be perfectly synchronised behind the second BC. This recreates an 45° polarised pulse (3), blocked by the second polariser (4). In the presence of an X-ray pulse (5), a phase-shift is introduced (6), preventing the perfect synchronisation of both PCs behind the second BC and creating an elliptical polarised pulse (7), partly transmitted through the second polariser (8).

schemes with 24 fs (RFS) and 23 fs (OS). The OS stabilises the timing jitter on the time scale of consecutive pulse-trains (10 Hz), while the intra-train (MHz) timing jitter is identical for both synchronisation schemes. In addition, different intra-train drift pattern were observed for both synchronisation schemes. The source of these drift pattern are yet to be investigated by the facility and open the possibility to reduce the timing jitter even more.

By comparing the experimentally measured arrival time spectra to a simulated arrival time spectrum, we were able to estimate the X-ray-induced refractive index. For X-ray photon energies of 9.3 keV and a peak fluence of 12 J/cm² in the center of the X-ray beam profile, the X-ray-induced refractive index change was $\Delta n_{\exp} = -5.7 \times 10^{-5}$. We showed that the commonly used Drude model overestimates the X-ray-induced refractive index change by nearly two orders of magnitude. We have demonstrated that the Maxwell-Garnett theory of dielectric mixtures, where we assume a dielectric mixture of a diamond host material and electronic polarons, describe the observed measurement very well. With the high versatility of this self-referenced timing tool, even in-situ arrival time measurements in a unstable free-flowing liquid jet are possible[37]. This can also be used to quantify the robustness of the Maxwell-Garnett model for various materials and liquids in the near future.

## Methods

### Common path interferometer (CPI)

The self-referenced detection scheme relies on the inherent stability and robustness of the CPI. The first idea for a CPI occurred in 1958, described by L. Mertz[38] and was applied a few years later to realise a Polarisation Fourier Spectrometer for astronomical measurements[39]. The advantage of this interferometer scheme is that the generated pulse-replica co-propagate through the setup, making it robust against environmental disturbances, e.g., mechanical vibrations or air flows. Any occurring disturbance affects both pulse-replica equally, thus, effectively cancelling out the distortion. The CPI setup built for this experiment is illustrated in Fig. 10. The main parts are two identical a-cut birefrigent crystals (BC) between two crossed polarisers. The transmission axis of the first polariser is set to 45°, such that behind the first polariser, the transmitted light has a defined linear polarisation of 45° (1). The 45° polarised pulse can equally be described as a linear combination of a horizontal (blue) and vertical (orange) polarisation components (PC) with equal amplitudes.

The first BC is aligned with its optical axis to be horizontal. Due to the birefringence of the crystal, the horizontal PC experiences the extraordinary refractive index $n_e$, while the vertical PC experiences the ordinary refractive index $n_o$ of the crystal. After passing the first BC, the two PCs are then time-delayed with respect to each other (2). The time-delay between the two PCs is defined by the ordinary $n_o$ and extraordinary $n_e$ refractive index and the crystal thickness.

The two PCs are transmitted through the (ideally) optically isotropic sample, experiencing the same refractive index. The second BC is rotated orthogonally to the first BC, such that the optical axis is now perpendicular to the optical axis of the first BC. The PC arriving first, which experienced the extraordinary refractive index $n_e$ in the first BC, now experiences the ordinary refractive index $n_o$ of the second BC, and vice versa for the trailing PC. In such a configuration, both PCs are temporally overlapped again behind the second BC. After the second BC, both PCs are effectively propagated through two crystals with identical thickness and accumulated the same amount of additional phase by experiencing the same effective refractive index. Both PCs are then exactly synchronised. The linear combination of both PCs recreates the original 45° linear polarised optical pulse (3).

This 45° polarised optical pulse is then completely blocked by the second polariser, whose transmission axis is oriented to be −45°. In this case, no optical pulse can be observed behind the second polariser (4).

If the amplitude or phase of one of the two PCs is artificially manipulated, the situation is different: In this case, the ultrashort X-ray pulse is temporally overlapped with the trailing vertical PC as shown in (5). The entire leading horizontal PC and half of the trailing vertical PC is transmitted through the sample before the X-ray pulse (green) strikes the sample. The X-ray pulse generates an electron cascade, modulating the electronic band structure, ultimately changing the complex refractive index of the sample. Due to this change of the refractive index, parts of the vertical PC, arriving after the X-ray pulse, pass through the sample with the modulated refractive index. These are then subject to a phase-shift of the transmitted optical light (6). Due to the chirp of the original optical pulse, the temporally leading red spectral parts of both PCs transmit through the sample without the modulated refractive index. The trailing spectrally blue parts of the leading horizontal PC also pass through the sample with the static refractive index, while the identical blue parts of the trailing horizontal PC passes through the sample with its X-ray-modulated refractive index and are phase-shifted by a small amount.

Behind the second BC, the original 45° polarisation can not be recreated by a linear combination of both synchronised PCs. The temporally leading lower wavelength parts of both PCs are still perfectly synchronised and recreate the original 45° polarisation, and are therefore blocked by the second polariser. However, the higher wavelength parts are not synchronised anymore due to the X-ray-induced phase-shift of these spectral parts in the trailing horizontal PC. The linear combination of these parts now yield an elliptically polarised optical pulse (7).

The perfectly synchronised first part of the optical pulse with the red spectral parts is still 45° polarised and still not transmitted by the second polariser. The elliptical polarised blue parts of the optical pulse can partially transmit through the second polariser, generating a self-referenced background-free arrival time signal (8). Depending on the relative arrival time between the optical and X-ray pulse, the spectral position of the transition between the original 45° polarisation to the elliptical polarisation changes for each optical / X-ray pulse pair and is clearly visible in the observed spectrum by a cut-off edge.

### X-ray-induced refractive index change

The electron-hole pair density after an X-ray pulse with known X-ray photon energy in a material with X-ray attenuation length $\mu$, can be calculated using

$$E_{abs} = E_0 \left( 1 - e^{-\frac{d}{\mu}} \right),  \qquad (4)$$

$$N = \frac{E_{abs}}{V} \frac{1}{E_{pair}}.  \qquad (5)$$

$E_{abs}$ is the total absorbed energy in a material with thickness $d$ of the incoming X-ray pulse with total energy $E_0$. The electron density $N$ is then calculated by dividing $E_{abs}$ by the X-ray irradiated volume $V$ and the electron-hole pair generation energy $E_{pair}$ which is 12.2 eV in diamond. In our experiment (9200 eV, 200 $\mu$J, 20 $\mu$m X-ray size, 50 $\mu$m thick diamond), the peak X-ray-induced electron density $N$ in the center of the Gaussian beam profile is calculated to be $4.7 \times 10^{19}$ electron-hole pairs cm$^{-3}$. The diamond properties used for the Drude and Maxwell-Garnett models are listed in Tables S1 and S2 (Supplementary Note 5), respectively.

The X-ray induced refractive index change is determined by

$$\Delta n(\omega) = n_A(\omega) - n_B(\omega),  \qquad (6)$$

where $n_B(\omega)$ and $n_A(\omega)$ are the real part of the refractive indices before (B) and after (A) the X-ray interaction with the diamond. Note that we use literature values of diamond for $n_B(\omega)$ (see Table S3 in Supplementary Note 5). The real part of the refractive index after the X-ray interaction is calculated by

$$n_A(\omega) = \sqrt{\frac{\sqrt{\mathrm{Re}[\epsilon(\omega)]^2 + \mathrm{Im}[\epsilon(\omega)]^2} + \mathrm{Re}[\epsilon(\omega)]}{2}},  \qquad (7)$$

with the complex dielectric function $\epsilon(\omega)$ for a given light frequency $\omega$. Here, $\epsilon_D(\omega)$ (Eq. (1)) and $\epsilon_{MG}(\omega)$ (Eq. (2)) are calculated and used to determine $n_A(\omega)$ (Eq. (7)) for the Drude and Maxwell-Garnett model, respectively.

### Linear propagation and expected self-referenced signal

The self-referenced arrival time signal is simulated by linear propagation of the original chirped laser pulse (centered around 400 nm) through each element of the CPI. The X-ray induced change of the electron density $N$ in the diamond sample is imprinted into the transmitted optical pulse by a change of the refractive index of the

diamond sample over a time span of tens of fs, depending on the electron cascading time[22, 23]. The linear propagation is described in detail in Supplementary Note 2.

The properties used for the simulation are compiled in Tab. S3 (Supplementary Note 5).

## Data availability
The data that support the findings of this study are available from the corresponding author(s) upon request.

## Code availability
The code for the simulation of the self-referenced arrival time spectra is available from the corresponding author(s) upon request.

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

## Acknowledgements
This work is supported by the Deutsche Forschungsgemeinschaft (DFG) via the Cluster of Excellence 'Advanced Imaging of Matter', EXC 2056, Project ID 390715994, via SFB925 ID 170620586 (TP A4), and by European XFEL. K.K. gratefully acknowledges funding by the DFG within the program "Sachbeihilfe" project ID 497431350 (KU 4184/1-1). S.S. and C.Br. acknowledge support from the European Cluster of Advanced Laser Light Sources (EUCALL) project which has received funding from the European Union's Horizon 2020 research and innovation programme under grant agreement No 654220. W.G. acknowledges partial funding from Narodowe Centrum Nauki through SONATA BIS 6 grant (2016/22/E/ST4/00543). The experiments were performed at the FXE instrument of European XFEL GmbH. We want to thank Jia Liu and the X-ray Diagnostics Group at European XFEL for helping out with a mount for the utilised Gotthard detector, and the European XFEL laser group for providing us with the optical laser beam. We thank Martin Knoll and Paul Frankenberger for vital setup tasks, which allowed this experiment.

## Author contributions
M.D, R.C., and A.G. conceived the experiment, and M.D. and A.G. coordinated the experiment. M.D. assembled the optical set-up. A.G. implemented the Gotthard detector into the DAQ and implemented on-the-fly analysis of the data. M.B., C.Bo., T.K.C., M.D., A.R.F., A.G., W.G., D.K., K.K., F.L., F.O., S.S., and P.Z. were involved in beamline control and performed the X-ray experiment. H.K. and M.T. developed the approach based on the Maxwell-Garnett theory and provided the related theoretical simulations. M.D. provided the Drude Simulation and analysed both simulations. M.D. analysed the data. M.D. and C.Br. interpreted the data. M.D. wrote the manuscript with extensive contributions from C.Br., H.K., M.T. and contribution from all other authors.

## Funding

## Competing interests
The authors declare no competing interests.
