## [Peer Review File · Nature Communications]

A sensitive high repetition rate arrival time monitor for X-ray Free Electron LasersREVIEWER COMMENTS

Reviewer #1 (Remarks to the Author):

In the manuscript entitled “A sensitive high repetition rate arrival time monitor for X-ray Free Electron Lasers”, written by Diez et al., the authors described a theoretical and experimental study on a new monitoring system to measure an arrival timing jitter between XFEL and optical pulses for high-repetition rate XFEL sources. The diagnostics of arrival timing is very important for optical-pump/X-ray probe experiments in a femtosecond resolution, because synchronization of two independent sources could have an ambiguity of the order of sub-ps. The arrival timing monitors have already been developed and used as indispensable tools in every XFEL facility, while they are operated for the XFEL machines with low-repetition rates (typically ~ 100 Hz). In this work, the authors tried to construct an arrival timing monitor that is compatible with a high-repetition rate machine (\sim MHz). In this case, one needs to analyze a very small change of the optical parameters under a weak interaction of an X-ray pulse to a sample. Otherwise, the sample could be seriously damaged. For this purpose, the authors used a diamond plate for the sample, and a subtle change of the refractive index was probed with a CPI that has high sensitivity and robustness. Using the system, they measured arrival timings for 1.1 MHz X-ray pulse trains.

I think this work is meaningful and important for ultrafast pump-probe experiments at a high-repetition XFEL source. The theoretical model based on the Maxwell-Garnett theory, rather than the Drude model that has been widely utilized for this purpose, looks appropriate. Still, I do not recommend this manuscript for publication in Nature Communications, because it does not include any conceptual leap nor substantial technological jumps. For example, the overall timing accuracy of 19 fs is comparable to values that have been already achieved with conventional methods. Also, I have some serious questions about the basic scheme of timing analysis, as shown the following comments (1-3). I believe a more specialized journal would be appropriate for this manuscript after major revisions.

1) In Fig. 4, the profiles with the positive and negative arrival times (for example, $t = -1500$ fs and $t = 1500$ fs) look mostly symmetric. However, the temporal response before/after the X-ray irradiation should be highly asymmetric, because the decay time of the free-electron cloud after the excitation is generally much longer than the leading time. Thus the existing arrival timing monitors (both spatial and spectral) are probing the leading edge of the X-ray pulse. However, your analysis in Fig. 4 is made not only for the leading edge (for the positive delays), but also for the terminating edge (for the negative delays) of the X-ray pulse. This is deeply puzzling me.

2) If the response time returning to the initial state is assumed to be very fast, you may use it for analysis. Even in this case, however, mechanism of the fast decay process should be clearly explained.

3) Also, the timing for the terminating edge does not necessarily correlates with the arrival timing, because a possible shot-to-shot variation of the temporal duration and intensity of X-ray pulse could provide serious complication in analysis.

4) In Fig. 6, a systematic change of the arrival timing for the intra bunches was observed for the RFS condition. Could you provide possible reasons ?

5) Please specify the temporal duration of the X-ray pulse.

Reviewer #2 (Remarks to the Author):

The paper by Diez et al. describes a method to measure the time arrival difference between FEL pulses and laser pulses at the FXE instrument of the EuXFEL.

They exploit the Maxwell-Garnett model to describe the refractive index changes in diamonds and show that this model is in much better agreement with the experimental data with respect to the most used Drude model.

The topic is quite relevant for the scientific community since laser pump-X-ray probe time-resolved experiments at free electron lasers are one of the most important tools in ultrafast science. Therefore, methods to accurately determine the delay between pump and probe pulses are important to increase the time resolution of the experiment and can help in collecting data in an efficient way.

The paper is properly written and the conclusions are widely supported by the experimental results and their modelization. The paper, including the supporting material, provides enough details for the work and data analysis to be reproduced. Nevertheless, I think the authors should clarify a few minor points in order for the paper to be published in Nature Communications.

1) As a general comment, I think it'd be beneficial, in order to make clear to which extent the described scheme can be used at other FEL sources, to discuss its expected operation range in terms of photon wavelength, pulse durations, and, in general, beam parameters (some more detailed questions about this in the following points).

2) When describing the Maxwell-Garnett model, the authors write that the free electrons created by the incoming X-ray pulses interact instantaneously with the bound electrons in the diamond. For sure the interaction is fast, but not instantaneous. The time scale of the interaction could become relevant when

dealing with ultra-short, even sub-femtosecond pulses. The authors should therefore provide an estimation of the interaction timescale and comment on this.

3) Still on the same topic, in the paragraph about “X-ray-induced Transient Refractive Index” the authors comment on the timescale of the electron cascades for different photon energies. Since these timescale are comparable with that of the FEL pulses, the authors should comment on the effect of different pulse-lengths on the accuracy of their method.

4) In the paper, they double the frequency of the 800 nm optical laser. What is the motivation for this choice? Moreover, the author should discuss in which optical wavelength range the described method would work.

5) How precise is the “more precise” optical synchronisation scheme?

6) The authors write that they can reach a 19 fs timing accuracy and that this precision is governed by the resolution of the spectrometer and can be increased by using a more dispersive grating.

7) They should make this last observation quantitative by estimating what the effect of a more (how much more would be reasonable?) dispersive grating would be on the timing accuracy.

8) Is there an explanation for the fact that the relative arrival times of the first 10 pulses within a pulse train exhibit a larger uncertainty?

9) It is quite reasonable that the non-perfect overlap between X-ray and optical pulses causes large arrival time amplitude variation. Still, it'd be important to provide an estimation of these variations. Moreover, the authors should discuss their choice of including only the highest 0.5%.

10) What is the error on the refractive index change -5.7×10^{-5} that comes from the fit to the experimental data?

Reviewer #3 (Remarks to the Author):

I have carefully read the manuscript entitled “A sensitive high repetition rate arrival time monitor for X-ray Free Electron Lasers” by M. Diez and co-workers.

In this manuscript the authors report the results of a pilot experiment performed at the European XFEL to offer an alternative approach to measure the jitter affecting the arrival time combining the X-Rays lased from the machine and optical laser pulses, delivered by a custom table-top light source, installed to offer to the users the possibility to implement time-resolved spectroscopies.

Such a problem is currently under investigation by more than one decade and is still far to be completely under control. The capability to perform a satisfactory beamtime at XFELs, with an appreciable S/N ratio, so taking advantage from all the machine's properties, is intrinsically dependent on these kind of studies.

My general comment is that this manuscript is interesting, well written and novel, from a technological point of view, and it offers a different approach to measure the arrival time between X-Rays and optical laser pulses. In my opinion it offers an alternative experimental approach to measure this jitter, if compared to previous methods. Indeed, implementing a scheme where i) the optical pulses are lying in the NIR/visible spectrum and ii) the sample is a well-known solid crystal, is effectively more convenient and less invasive than experimental layouts implementing noble gas targets (sample), and/or THz radiation used as optical streaking field. Of course, THz-based streaking at FELs can offer stronger feedback, being able to reveal not only the arrival time of the X-Rays, but their temporal profile, too.

The phenomenological model used to extrapolate the variation of the index of refraction is well discussed (“X-ray-induced Refractive Index Change”), presenting two distinct models, as the well known Drude approach, which has some intrinsic limits to properly predict the observed results, particularly overestimated the intensity of the variation of the index of refraction and self-modulation effects, too. The additional Maxwell-Garnett model helps the reader to realize the limits of the Drude one and offers a novel and more robust phenomenological approach to explain these kind of data.

I suggest to the authors to address few minor changes:

- 1) In the “Experimental details” section please briefly mention the main properties of the BBO used to frequency double the incoming NIR radiation (thickness, cut angle, efficiency), and reports this also in the final “Methods” section.

2) In the “Experimental details” section please briefly mention in more detail the properties of the custom table-top light source. If this system has been presented and described in some technical publication, please mention it in the references.

3) In the “Results” section the authors mentioned that the optical transport is composed by three different mirrors, introducing a not negligible pointing instability. Can the authors discuss the properties of these mirrors as their grazing angle, coating(s), a comment about the thermal loading (if exist), the source jitter (instabilities at the X-Rays emission point), mechanical/seismic noise of the mirrors (if known) and a rough value about the mentioned pointing instability (I suppose in the experimental station focal plane), measured as microns of instabilities on the focal plane and microradians. How do this pointing instability affect the measurement presented in the manuscript? Can the authors comment and quantify this effect?

4) As mentioned above, this approach is robust and less invasive than, for example, THz-based streaking layouts. On the other hand, it is still not useful to get a proper temporal profile of the X-Rays delivered from the machine. Can the authors discuss this point in the “Conclusions” section to overview both pros and cons of their approach.

Few additional suggestions for the authors:

1) I think that Figures 2 and 3 can be merged into one single panel, more compact and elegant. Please, rescale a bit both horizontal and vertical labels and use scientific notations for the vertical ones. I think that a different colormap can be used, too. This will help the visibility of the difference between the Drude and Maxwell-Garnett models.

2) The caption of Figure 3 holds three different typographical errors: please insert a space between the numerical value of the beam diameter and “ μm ”.

3) I think that Figures 5 and 6 can also be merged and re-organized into a single panel.

4) The size of the labels can be applied to Figures 5/6/7/8 and 9. Please, rescale a bit both horizontal and vertical labels and use scientific notations for the vertical ones.

5) Figure S6 should be slightly changed. Different colored slices are one above each other (yellow and orange ones). The Delta t labels should be reduced, too, to not overlap the experimental lines of the calibration curves.

Summarizing, I suggest accepting this manuscript for publication in Nat. Communications after minor changes.

We thank the reviewers for their remarks, which we treated in order to improve the quality of this article. Below we repeat each reviewer remark (in bold) and deliver our answer including changes made to the manuscript, one by one:

Reviewer #1 (Remarks to the Author):

In the manuscript entitled “A sensitive high repetition rate arrival time monitor for X-ray Free Electron Lasers”, written by Diez et al., the authors described a theoretical and experimental study on a new monitoring system to measure an arrival timing jitter between XFEL and optical pulses for high-repetition rate XFEL sources. The diagnostics of arrival timing is very important for optical-pump/X-ray probe experiments in a femtosecond resolution, because synchronization of two independent sources could have an ambiguity of the order of sub-ps. The arrival timing monitors have already been developed and used as indispensable tools in every XFEL facility, while they are operated for the XFEL machines with low-repetition rates (typically ~100 Hz). In this work, the authors tried to construct an arrival timing monitor that is compatible with a high-repetition rate machine (~MHz). In this case, one needs to analyze a very small change of the optical parameters under a weak interaction of an X-ray pulse to a sample. Otherwise, the sample could be seriously damaged. For this purpose, the authors used a diamond plate for the sample, and a subtle change of the refractive index was probed with a CPI that has high sensitivity and robustness. Using the system, they measured arrival timings for 1.1 MHz X-ray pulse trains.

I think this work is meaningful and important for ultrafast pump-probe experiments at a high-repetition XFEL source. The theoretical model based on the Maxwell-Garnett theory, rather than the Drude model that has been widely utilized for this purpose, looks appropriate. Still, I do not recommend this manuscript for publication in Nature Communications, because it does not include any conceptual leap nor substantial technological jumps. For example, the overall timing accuracy of 19 fs is comparable to values that have been already achieved with conventional methods. Also, I have some serious questions about the basic scheme of timing analysis, as shown the following comments (1-3). I believe a more specialized journal would be appropriate for this manuscript after major revisions.

We thank the reviewer for this concise summary. We hope to convince her/him that this work not only represents a technological leap but also conceptual one.

However, we do disagree with the opinion that our approach has no conceptual and technological novelties compared to the widely used conventional methods. While our scheme does not achieve more precise timing information it is the only one which can be easily used at high repetition rate XFEL facilities and with intense mJ strong X-ray pulses ranging from soft to hard X-ray photon energies. With intense X-ray pulses at high repetition rates, conventional timing tool samples are either destroyed (e.g., the known heat pile-up after several pulses in Si₃N₄ or comparable samples) or do not return to their initial state within the required 10-100 ns time scale. Other successful timing tool methods such as THz streaking are not as universally applicable over a wide range of X-ray photon energies and are experimentally complicated to set-up.

1) In Fig. 4, the profiles with the positive and negative arrival times (for example, $t=-1500$ fs and $t=1500$ fs) look mostly symmetric. However, the temporal response before/after the X-ray irradiation should be highly asymmetric, because the decay time of the free-electron cloud after the excitation is generally much longer than the leading time. Thus the existing arrival timing monitors (both spatial and spectral) are probing the leading edge of the X-ray pulse. However, your analysis in Fig. 4 is made not only for the leading edge (for the positive delays), but also for the terminating edge (for the negative delays) of the X-ray pulse. This is deeply puzzling me.

The generated 'free-electrons' in the diamond sample have a much longer lifetime (around 1-3 ns, see e.g. Diamond Films and Techn. 8(5), 369 (1998)) than the exciting X-ray pulse and the simultaneously probing chirped optical pulse with its two polarisation components. Therefore, after the initial generation of the free-electron density by the X-ray pulse, this electron density remains basically unchanged over the entire measurement window of just a few ps.

For clarity, we added the following sentence to the manuscript in the "X-ray Induced Transient Refractive Index" section section:

"For high-energy photons (>10 keV), the cascading can reach time scales of 100 femtoseconds [23]. The so generated electrons in the conduction band have a long lifetime in the 1-3 ns range [24]."

As the reviewer said, the temporal response before and after the X-ray pulse excitation is highly asymmetric. We tried to describe the emergence of the symmetric arrival-time signal in Figure 4 in the Methods section (CPI). Note that the leading and trailing edge (terminating edge in the words of the reviewer) of the arrival-time signal is not generated by the leading and trailing edge of the X-ray pulse. The entire X-ray pulse generates the free-electron density in the material, which requires between 10 to 100 fs, depending on X-ray pulse length and X-ray photon energy (cascading time higher for higher X-ray photon energies). Due to the small separation of only a few ps of the two optical polarization components, both polarization components experience nearly the same electron density of the diamond. Therefore, all parts of the two polarization components which are transmitted through the diamond *before* the X-ray pulse arrives are identical. Due to the time shear between both polarization components there is a spectral region which was transmitted in the leading polarization component before the X-ray pulse arrived, but the same spectral region of the trailing polarization component was transmitted after the X-ray pulse arrived. This spectral region of both polarization components (of the chirped optical pulse) is not identical anymore, due to the X-ray-induced refractive index change for the trailing polarization component. Hence, this spectral region can not recreate the original 45° polarization behind the second birefringent crystal, and therefore, is partially transmitted through the second polarizer.

The parts of both polarization components which are transmitted through the diamond after the X-ray pulse arrives are identical again and can vanish behind the second polarizer. Our manuscript should contain all of the above explanations, but we changed the paragraph (marked-up on p. 12) for enhanced clarity:

"The temporally leading lower wavelength parts of both PCs are still perfectly synchronised and recreate the original 45° polarisation, and are therefore blocked by the second polariser. However, the higher wavelength parts are not synchronised anymore due to the

X-ray-induced phase-shift of these spectral parts in the trailing horizontal PC. The linear combination of these parts now yield an elliptically polarized optical pulse (7)".

2) If the response time returning to the initial state is assumed to be very fast, you may use it for analysis. Even in this case, however, mechanism of the fast decay process should be clearly explained.

The lifetime of the excited carrier is far longer (order of 1-3 ns) than all chirped pulses involved (see previous point) and written in the manuscript, so it remains certainly smaller than about 100 ns, as required for the MHz repetition rate requirement.

3) Also, the timing for the terminating edge does not necessarily correlate with the arrival timing, because a possible shot-to-shot variation of the temporal duration and intensity of X-ray pulse could provide serious complication in analysis.

See above, we are not sensitive to the temporal rising and falling edges of the X-ray pulse, only to the total electron density it created via an electron cascade in the timing tool material. The spectral distance between the rising and falling edges of the arrival time spectra are solely defined by the temporal separation of the two polarization components. Since we use a common path interferometer this temporal separation is very stable. X-ray pulse intensity fluctuations are only affecting the X-ray-induced refractive index change and therefore the amplitude of the arrival time spectra. As described in the manuscript we can analyze all spectra with a SNR better than 2.

4) In Fig. 6, a systematic change of the arrival timing for the intra bunches was observed for the RFS condition. Could you provide possible reasons ?

This is a good question: The intra and inter train feedback system act on different time scales and with different drifts. Currently there is no explanation for the detailed behavior, e.g., from train to train, and remains outside of the scope of this article.

5) Please specify the temporal duration of the X-ray pulse.

The pulse width for hard x-ray pulses has not yet been measured at European XFEL. We added the following information in the section "Experimental details":

"The experiment was carried out at the FXE instrument at the end of the SASE1 photon beamline of EuXFEL at a fixed X-ray photon energy of 9.3 keV and a mean X-ray pulse energy of 300 μ J. The pulse width is expected to be around 50 fs (FWHM) (W. Decking et al., NAture PHotoNics | VOL 14 | June 2020 | 391–397)."

Reviewer #2 (Remarks to the Author):

The paper by Diez et al. describes a method to measure the time arrival difference between FEL pulses and laser pulses at the FXE instrument of the EuXFEL. They exploit the Maxwell-Garnett model to describe the refractive index changes in diamonds and show that this model is in much better agreement with the experimental data with respect to the most used Drude model.

The topic is quite relevant for the scientific community since laser pump-X-ray probe time-resolved experiments at free electron lasers are one of the most important tools in ultrafast science. Therefore, methods to accurately determine the delay between pump and probe pulses are important to increase the time resolution of the experiment and can help in collecting data in an efficient way.

The paper is properly written and the conclusions are widely supported by the experimental results and their modelization. The paper, including the supporting material, provides enough details for the work and data analysis to be reproduced. Nevertheless, I think the authors should clarify a few minor points in order for the paper to be published in Nature Communications.

We thank the reviewer for this concise summary.

1) As a general comment, I think it'd be beneficial, in order to make clear to which extent the described scheme can be used at other FEL sources, to discuss its expected operation range in terms of photon wavelength, pulse durations, and, in general, beam parameters (some more detailed questions about this in the following points).

The scheme should work fine with soft and hard X-ray sources, but one needs to be cautious with soft X-rays, since the absorption cross section increases and diamond could be easier destroyed. The pulse width is not important, as long as it is shorter than the timing jitter of a few hundred femtoseconds. In general we can easily work with the same beam parameters as known from spectral and spatial encoding schemes, and now with even less X-ray pulse intensity, since the self-referenced scheme is more sensitive. A thorough analysis for each different beam parameter is underway, but outside the scope of this article. We added the following explanation in the revised introduction:

“In this letter we describe a new self-referenced timing-tool scheme with an increased sensitivity. This scheme could be applied at all x-ray wavelengths from soft (<1keV) to very hard (>25keV) X-radiation.”

2) When describing the Maxwell-Garnett model, the authors write that the free electrons created by the incoming X-ray pulses interact instantaneously with the bound electrons in the diamond. For sure the interaction is fast, but not instantaneous. The time scale of the interaction could become relevant when dealing with ultra-short, even sub-femtosecond pulses. The authors should therefore provide an estimation of the interaction timescale and comment on this.

The X-ray pulse creates energetic free charges in the sample, which subsequently cascade to form many conduction-band electrons. These are then probed with optical laser pulses that are intentionally chirped and thus much longer than the interaction time between free and bound electrons. However, the dominant timescale for the change of the refractive index is the cascading time for creating free electrons (10-100 fs) after the X-ray pulse strikes the sample. The bound electrons respond to the free electrons on a much faster timescale (~0.3fs) as, e.g., deduced from spectroscopic measurements of the dielectric function in the optical limit (i.e., from the position of the maximum of the (absorptive) imaginary part of the

dielectric function, see K. Ramakrishna, J. Vorberger, J.Phys.: Condes. Matter **32** (2020) 095401). We clarified the term “instantaneous” in the ms:

“The free electrons created by the incoming X-ray pulses interact instantaneously (i.e. within 0.3 fs) with the bound electrons in the diamond crystal, distort their equilibrium distribution and thus polarise the diamond lattice (reference).”

3) Still on the same topic, in the paragraph about “X-ray-induced Transient Refractive Index” the authors comment on the timescale of the electron cascades for different photon energies. Since these timescales are comparable with that of the FEL pulses, the authors should comment on the effect of different pulse-lengths on the accuracy of their method.

If the X-ray pulse is much shorter than the cascading time, then the timing accuracy is not affected, as stated in the manuscript: “The free electrons created by the incoming X-ray pulses interact instantaneously (i.e. within 0.3 fs) with the bound electrons in the diamond crystal, distort their equilibrium distribution and thus polarise the diamond lattice”.

If the X-ray pulse is very long in comparison to the electron cascading time, the X-ray pulse is continuously generating high energy electrons which are then cascading to their final energy distribution. The total time of the transient refractive index change is defined by the temporal shape of the X-ray pulse and the constant electron cascading time (low photon energy fast cascading time ~10fs, high photon energy photons ~100fs) for each absorbed X-ray photon. The fastest refractive index change is on the time scale of the electron cascading time, assuming the X-ray pulse is instantaneous. Thus, longer X-ray pulses increase the duration of the X-ray induced refractive index change, which in turn determines the slope of the rising and falling edge of the arrival-time spectrum. For longer X-ray pulses the slope of the leading and trailing edges of the arrival time spectrum is smaller and therefore the fitting algorithm described in the Methods section becomes accordingly less precise.

As a side note, if the X-ray pulses are much longer than 100 fs then the intrinsic timing jitter of the machine would be shorter than the X-ray pulse duration itself, obviating the necessity for a timing tool.

4) In the paper, they double the frequency of the 800 nm optical laser. What is the motivation for this choice? Moreover, the author should discuss in which optical wavelength range the described method would work.

The X-ray induced refractive index change for the 800 nm optical region and the 400 nm optical region is nearly identical. Thus, according to Eq.3 of the manuscript, the X-ray induced phase-shift is doubled when changing from 800 nm to 400 nm, enhancing the signal strength (amplitude) of the arrival time spectrum. We would therefore expect only a modest wavelength-dependence, but this would have to be tested experimentally.

5) How precise is the “more precise” optical synchronisation scheme?

This was measured, see e.g. Fig. 5 of the manuscript, and delivered 154 fs (FWHM) for the RF and 83 fs (FWHM) for the optical synchronization scheme, as measured over a period of a few min. We also noted in the manuscript:

“One should note that over a longer time periods, the RF-synchronisation timing jitter is expected to become worse, while the optical synchronisation timing jitter is expected to remain within the 83 fs FWHM window.”

6) The authors write that they can reach a 19 fs timing accuracy and that this precision is governed by the resolution of the spectrometer and can be increased by using a more dispersive grating. They should make this last observation quantitative by estimating what the effect of a more (how much more would be reasonable?) dispersive grating would be on the timing accuracy.

We used a 600 l/mm grating (we added this information into the manuscript). Increasing the groove density to e.g. 1200 l/mm would deliver an increased timing accuracy by a factor of 2, thus to less than 10 fs. We added the info in the following sentence in the Results section: “This precision is governed by the utilised spectrometer resolution and can be increased by using a more dispersive , e.g., by about a factor of two when using a 1200 l/mm.”

7) Is there an explanation for the fact that the relative arrival times of the first 10 pulses within a pulse train exhibit a larger uncertainty?

This is likely due to the electron bunch feedback control as implemented within the machine control system. Added remark: Not only the arrival time is affected, but also the bunch energy and pointing of the pulses are affected. Meanwhile (after this experiment) the feedback control diagnostics have been significantly improved.

This is also mentioned in the text with a corresponding source with more details.

“Analysis of the X-ray pointing jitter and the rejection of weak arrival-time spectral intensities are given in reference (33)”

8) It is quite reasonable that the non-perfect overlap between X-ray and optical pulses causes large arrival time amplitude variation. Still, it'd be important to provide an estimation of these variations. Moreover, the authors should discuss their choice of including only the highest 0.5%.

The timing results shown in Figs. 5 and 6 used actually all signals with a S/N ratio larger than two, corresponding to about 60 % of all shots, thus the variation in arrival time versus signal amplitude as reported in the manuscript contains this larger distribution of arrival time amplitude variation. Meanwhile EuXFEL has much larger hard x-ray pulse intensities in the 2-4 mJ range, which nicely surpass the lower intensities in this article.

For the quantitative analysis of the timing signal and its relation to the refractive index we selected more than 1000 traces (1600, as outlined in the manuscript) with the highest signal amplitudes (see Fig. 7, the top 0.5 % signals), and these corresponded to about 0.5 % of the data traces. This rather strict limitation ensures that spatial overlap is ensured, since the top 0.5 % signal amplitudes in the arrival time spectra also contained the top 0.5 % of x-ray pulse intensities (0.68 mJ).

We added these informations in the main text on p. 9:

“We use the highest 0.5% (1600 measurements) of the self-referenced spectra amplitudes to ensure that i) the X-ray pump and laser probe pulses fully overlap, and ii) a well-known (largest) X-ray pulse energy (here: 0.68 mJ) strike the sample.”

As a side remark, below is a single spectrum with a low signal to noise of about 2 (from reference [34]), which we set as a lower limit to measure the arrival time. Using the simulation described in the manuscript, the refractive index change to generate such a spectral amplitude is -1×10^{-5} , corresponding to a X-ray pulse energy of 90 uJ, assuming perfect overlap and the reported focus size of the X-ray beam.

Figure 7.19.: X-ray arrival-time spectrum (blue) with the smallest required amplitude to fulfill a SNR greater than 2. The noise floor is indicated by the blue shaded area. The simulation reproducing the experimental spectrum is shown in orange. The error bars are indicated by the orange shaded area. The leakage of the original optical pulse through the crossed polarizer setup can be recognized and is implemented in the simulation and indicated by the purple dashed line.

Figure adopted from “Diez, M. A Self-Referenced Timing-Tool for High Repetition X-ray Free-Electron Laser Sources. (Ph.D. thesis, University of Hamburg, 2022)”

9) What is the error on the refractive index change -5.7×10^{-5} that comes from the fit to the experimental data?

In the manuscript we averaged the shown arrival time spectra (orange) to generate the red reference experimental arrival time spectrum (Fig. 7). The amplitude of the simulation was fitted to match the amplitude of this reference experimental spectrum, hence we cannot determine an error. However, we analysed every amplitude of each single arrival time spectrum and evaluated the corresponding refractive index change. The mean value of the refractive index of all orange arrival time spectra in Fig. 7 is -5.7×10^{-5} with a standard deviation of 1.7×10^{-6} .

Reviewer #3 (Remarks to the Author):

I have carefully read the manuscript entitled “A sensitive high repetition rate arrival time monitor for X-ray Free Electron Lasers” by M. Diez and co-workers.

In this manuscript the authors report the results of a pilot experiment performed at the European XFEL to offer an alternative approach to measure the jitter affecting the arrival time combining the X-Rays lased from the machine and optical laser pulses, delivered by a custom table-top light source, installed to offer to the users the possibility to implement time-resolved spectroscopies.

Such a problem is currently under investigation by more than one decade and is still far to be completely under control. The capability to perform a satisfactory beamtime at XFELs, with an appreciable S/N ratio, so taking advantage from all the machine's properties, is intrinsically dependent on these kind of studies.

My general comment is that this manuscript is interesting, well written and novel, from a technological point of view, and it offers a different approach to measure the arrival time between X-Rays and optical laser pulses. In my opinion it offers an alternative experimental approach to measure this jitter, if compared to previous methods. Indeed, implementing a scheme where i) the optical pulses are lying in the NIR/visible spectrum and ii) the sample is a well-known solid crystal, is effectively more convenient and less invasive than experimental layouts implementing noble gas targets (sample), and/or THz radiation used as optical streaking field. Of course, THz-based streaking at FELs can offer stronger feedback, being able to reveal not only the arrival time of the X-Rays, but their temporal profile, too.

The phenomenological model used to extrapolate the variation of the index of refraction is well discussed ("X-ray-induced Refractive Index Change"), presenting two distinct models, as the well known Drude approach, which has some intrinsic limits to properly predict the observed results, particularly overestimated the intensity of the variation of the index of refraction and self-modulation effects, too. The additional Maxwell-Garnett model helps the reader to realize the limits of the Drude one and offers a novel and more robust phenomenological approach to explain these kind of data.

We thank the reviewer for this concise summary. We also acknowledge the THz based streaking tool remark, which we also mention in the manuscript.

I suggest to the authors to address few minor changes:

1) In the "Experimental details" section please briefly mention the main properties of the BBO used to frequency double the incoming NIR radiation (thickness, cut angle, efficiency), and reports this also in the final "Methods" section.

We added the following information into the experimental details section :

"The 800~nm fundamental optical beam was frequency doubled with a beta-BBO (thickness of 0.5mm $\theta = 29.2$ deg and $\phi = 90$ deg) to generate optical pulses centered around 400~nm with a FWHM bandwidth of around 20~nm with a conversion efficiency of around 15%."

2) In the "Experimental details" section please briefly mention in more detail the properties of the custom table-top light source. If this system has been presented and described in some technical publication, please mention it in the references.

We have mentioned the appropriate reference about the custom-made laser source is already in the manuscript: "The in-house developed EuXFEL optical pump-probe laser system was utilized, which is synchronised to the facility's main oscillator, and matches any chosen X-ray pulse pattern of the facility [16]". The details therein go beyond the scope of this article, so we restricted the text details to the key laser parameters.

3) In the “Results” section the authors mentioned that the optical transport is composed by three different mirrors, introducing a not negligible pointing instability. Can the authors discuss the properties of these mirrors as their grazing angle, coating(s), a comment about the thermal loading (if exist), the source jitter (instabilities at the X-Rays emission point), mechanical/seismic noise of the mirrors (if known) and a rough value about the mentioned pointing instability (I suppose in the experimental station focal plane), measured as microns of instabilities on the focal plane and microradians. How do this pointing instability affect the measurement presented in the manuscript? Can the authors comment and quantify this effect?

For the X-Ray beamline properties where the beam transport is described, we added the related reference “H. Sinn et al., J. Synchr. Rad. 26, 692” to the existing sentence: “During the experiment, the X-ray beam pointing was not stable due to subtle mechanical vibrations on the mirror system (comprising three mirrors) in the photon delivery tunnel, located several hundreds of meters upstream from the sample position [33]”.

The pointing instabilities have been mentioned in the manuscript together with a reference [34], where further analysis and details are provided, see e.g.: “Analysis of the X-ray pointing jitter and the rejection of weak arrival-time spectral intensities are given in reference [34]” (p. 7). This reference also contains the following figure showing the pointing instabilities:

Figure 7.9.: X-ray pulse statistics of over 160 000 X-ray pulses. The spatial jitter, measured at the IPM position at the FXE beamline, is shown in a). The average distance of each pulse number within a pulse train to the COG is shown in b). The average pulse energy of the pulses within a pulse train is illustrated in c) for all pulses (blue), only the X-ray pulses with a spatial jitter less than 0.1 mm from the COG (orange) and with a spatial jitter more than 0.1 mm from the COG.

Figure adopted from “Diez, M. A Self-Referenced Timing-Tool for High Repetition X-ray Free-Electron Laser Sources. (Ph.D. thesis, University of Hamburg, 2022)”

- 4) As mentioned above, this approach is robust and less invasive than, for example, THz-based streaking layouts. On the other hand, it is still not useful to get a proper temporal profile of the X-Rays delivered from the machine. Can the authors discuss this point in the “Conclusions” section to overview both pro and cons of their approach.**

We added the following sentence to the Conclusion:

“In addition, the presented method possesses some advantages over other commonly used timing tool schemes. It is easier to set up and operate than a THz-Streaking experiment (Grgura2012) and it is more sensitive and less invasive than the established spatial- and spectral encoding schemes.”

Few additional suggestions for the authors:

- 1) I think that Figures 2 and 3 can be merged into one single panel, more compact and elegant. Please, rescale a bit both horizontal and vertical labels and use scientific notations for the vertical ones. I think that a different colormap can be used, too. This will help the visibility of the difference between the Drude and Maxwell-Garnett models.**

Thank you for this suggestion. We already had this fused graph (figs 2 and 3 together) in an earlier manuscript version, but this made the figure rather large and the caption getting overfilled. We hope that the current (=unchanged) split version serves the reader for easier readability.

- 2) The caption of Figure3 holds three different typographical errors: please insert a space between the numerical value of the beam diameter and “ μm ”.**

Thanks, done.

- 3) I think that Figures 5 and 6 can also be merged and re-organized into a single panel.**

Thank you for this suggestion. We also thought about merging these two figures before submitting the manuscript, but it proved difficult to make it visually pleasing with the different x-axes, which appears better in the current version with two separated figures.

- 4) The size of the labels can be applied to Figures 5/6/7/8 and 9. Please, rescale a bit both horizontal and vertical labels and use scientific notations for the vertical ones.**

Thank you, we tweaked the labels in figures 5 and 6 for improved visibility.

- 5) Figure S6 should be slightly changed. Different colored slices are one above each other (yellow and orange ones). The Δt labels should be reduced, too, to not overlap the experimental lines of the calibration curves.**

We changed the Δt labels, but kept the color scheme.

Summarizing, I suggest accepting this manuscript for publication in Nat. Communications after minor changes.

We thank all reviewers for their thoughtful and good remarks, and made changes accordingly. We hope the manuscript is now in a good state for publication.

REVIEWERS' COMMENTS

Reviewer #1 (Remarks to the Author):

I would appreciate the authors for providing the revised manuscript and the reply letter to our questions. Thanks to those materials, I think I am now able to understand the concept correctly, and to evaluate the novelty of the method appropriately. I am happy to support the acceptance of the article in Nature Communications.

One additional comment: It would be nice if you could refer a typical value of the phase sensitivity of the conventional method, such as the spectral encoding technique. This will enforce the advantage of the excellent phase sensitivity ($5.7e-5$) of the present method.

Reviewer #2 (Remarks to the Author):

The authors addressed all my concerns and modified the paper accordingly. In my opinion, it fully deserves publication in Nature Communication.

Reviewer #3 (Remarks to the Author):

The current state of the manuscript entitled "A sensitive high repetition rate arrival time monitor for X-ray Free Electron Lasers", by M. Diez and co-workers, is properly revised.

The authors properly to all of my previous request for minor changes. The manuscript is well-written and organized and present an exhaustive set of data, corroborated by a phenomenological model based on the Maxwell-Garnett theory. The authors emphasize and discuss why such a theory is more suitable than the Drude one, too.

In my opinion, this experimental approach is extremely interesting since it presents a novel, compact, self-referenced and not-invasive scheme to characterize x-ray pulses at large scale facilities as XFELs. It could be a starting point for more advanced schemes, not necessary for scientists focused on diagnostic problems, but for the entire communities performing experiments at XFELs. I recommend the publication in Nat. Comm. in its current form.

Response to Reviewers

Reviewer #1 (Remarks to the Author):

I would appreciate the authors for providing the revised manuscript and the reply letter to our questions. Thanks to those materials, I think I am now able to understand the concept correctly, and to evaluate the novelty of the method appropriately. I am happy to support the acceptance of the article in Nature Communications.

One additional comment: It would be nice if you could refer a typical value of the phase sensitivity of the conventional method, such as the spectral encoding technique. This will enforce the advantage of the excellent phase sensitivity ($5.7e-5$) of the present method.

We thank the reviewer for this valuable suggestion. We had already been aware of this important point and measured the absorption in the spectral encoding scheme back to back with our self-referenced scheme. The spectral encoding data are shown in the figures below (Fig. S7: spectral encoding scheme with an invasive YAG crystal, Fig. S8: spectral encoding scheme with diamond. The horizontal axis in pixels corresponds to wavelength of the chirped laser pulse). While we observed a timing tool signal with the YAG material we could not detect any absorption signal with the spectral encoding scheme in diamond, and can only conclude that its phase sensitivity is not measurable in diamond samples. The physical reason for this behavior is that x-rays produce orders of magnitude less conduction band electrons in diamond than in e.g. YAG. This also illustrates the superiority of our phase-sensitive timing tool.

We have added a section in the Supplemental Material illustrating this point.

Figure S7. Spectral encoding in YAG. The difference signals of a spectral encoding test measurement are shown in the 2-dimensional plot. The relative X-ray arrival time is systematically scanned from early (top) to later (bottom) arrival-times. Four example measurements are extracted on the right-hand side. The actual spectral encoding spectrum is shown in blue, the reference spectrum shown in light blue, and the difference signal with the imprinted X-ray arrival-time is shown in orange.

Figure S8. Difference spectra of a spectral encoding time-delay scan in diamond. Difference signals of the entire measurements are shown in the left 2-dimensional image. On the four sub-panels on the right, four exemplary measurements are shown with the spectral encoding spectrum (blue), reference spectrum (light blue) and difference signal in orange. No X-ray-induced spectral encoding signal can be observed.

Reviewer #2 (Remarks to the Author):

The authors addressed all my concerns and modified the paper accordingly. In my opinion, it fully deserves publication in Nature Communication.

Reviewer #3 (Remarks to the Author):

The current state of the manuscript entitled "A sensitive high repetition rate arrival time monitor

for X-ray Free Electron Lasers", by M. Diez and co-workers, is properly revised.

The authors properly to all of my previous request for minor changes. The manuscript is well-written and organized and present an exhaustive set of data, corroborated by a phenomenological model based on the Maxwell-Garnett theory. The authors emphasize and discuss why such a theory is more suitable than the Drude one, too.

In my opinion, this experimental approach is extremely interesting since it presents a novel, compact, self-referenced and not-invasive scheme to characterize x-ray pulses at large scale facilities as XFELs. It could be a starting point for more advanced schemes, not necessary for scientists focused on diagnostic problems, but for the entire communities performing experiments at XFELs. I recommend the publication in Nat. Comm. in its current form.

We thank all three referees for their thoughtful comments and final assessment. We are content that all referees find this contribution worthy of publication in Nature Communications.